# IEUBK Modeling of Children’s Blood Lead Levels in Homes Served by Private Domestic Wells in Three Illinois Counties

**DOI:** 10.3390/ijerph21030337

**Published:** 2024-03-13

**Authors:** Sarah Keeley, Samuel Dorevitch, Walton Kelly, David E. Jacobs, Sarah D. Geiger

**Affiliations:** 1Department of Occupational & Environmental Health Sciences, School of Public Health, West Virginia University, Morgantown, WV 26506, USA; 2Division of Environmental and Occupational Health Sciences, School of Public Health, University of Illinois Chicago, Chicago, IL 60612, USA; sdorevit@uic.edu (S.D.); djacobs@nchh.org (D.E.J.); 3Institute for Environmental Science and Policy, University of Illinois Chicago, Chicago, IL 60612, USA; 4Illinois State Water Survey, Prairie Research Institute, University of Illinois Urbana-Champaign, Champaign, IL 61820, USA; wkelly@illinois.edu; 5National Center for Healthy Housing, Columbia, MD 21044, USA; 6Department of Kinesiology and Community Health, University of Illinois Urbana-Champaign, Champaign, IL 61820, USA; smurphy7@illinois.edu

**Keywords:** epidemiology, health disparities, social determinants of health, lead poisoning, well water

## Abstract

Lead is known to impair neurocognitive development in children. Drinking water is routinely monitored for lead content in municipal systems, but private well owners are not required to test for lead. The lack of testing poses a risk of lead exposure and resulting health effects to rural children. In three Illinois counties, we conducted a cross-sectional study (n = 151 homes) examining water lead levels (WLLs), water consumption, and water treatment status to assess risk of lead exposure among residents using private water wells. Since blood lead levels (BLLs) were not available, EPA’s Integrated Exposure Uptake Biokinetic (IEUBK) modeling was used to estimate the incremental contribution of WLL to BLL, holding all other sources of lead at their default values. Nearly half (48.3%) of stagnant water samples contained measurable lead ranging from 0.79 to 76.2 µg/L (median= 0.537 µg/L). IEUBK modeling showed BLLs rose from 0.3 to 0.4 µg/dL when WLLs rose from 0.54 µg/L (the tenth percentile) to 4.88 µg/L (the 90th percentile). Based on IEUBK modeling, 18% of children with a WLL at the 10th percentile would have a BLL above 3.5 µg/dL compared to 27.4% of those with a WLL at the 90th percentile. These findings suggest that the consumption of unfiltered well water likely results in increased blood lead levels in children.

## 1. Introduction

Lead (Pb) exposure is associated with negative neurocognitive health effects in children on memory, gross motor skills, attention, and IQ [1,2,3,4,5] as well as physical effects such as stunted growth [6]. Studies have shown that even low levels of water lead levels (WLLs) (3.3 µg/L or greater) are associated with elevated blood lead levels (BLLs) in children 1–5 years old (adjusted odds ratio [OR] 4.7, 95% confidence interval [CI] 2.1–10.02) [7]. In Flint, Michigan, the increase in WLLs led to an increase in the frequency of BLLs >5 µg/dL among children, from 3% to 5% [8]. Drinking water is estimated to account for up to 20% of total Pb exposure in the U.S., but the risk is higher for infants who rely on formula; in fact, water can account for 45–60% of infants’ daily Pb exposure [9]. Community water supply systems are regulated by the EPA Lead and Copper Rule, which specifies a current action level of 15 µg/L. If WLLs exceed the action level, actions are required to reduce exposures to water Pb [10]. Owners of private domestic wells (PDWs) must measure WLLs for real estate transactions only in Rhode Island [11]. In other states, WLL testing is voluntary and infrequent [10]. Studies have shown that children who rely on PDWs are more likely to have elevated BLLs. A recent study found that North Carolina children on PDWs were 20% more likely to have an elevated BLL than children on community water sources [12]. Despite this information, there is still very little data on the association between BLLs and WLLs of children who rely on PDWs. Over 13 million private wells in the U.S. are the primary source of drinking water for an estimated 44 million people [13,14]. Thus, children who live in homes with PDW water may be at risk for ingestion of Pb, but because of the lack of water Pb testing, the scale of this public health threat is not known.

Pb in drinking water from PDWs is especially concerning in rural areas of the U.S. A study combining U.S. Geologic Survey and U.S. census data showed PDWs are more prevalent in areas of low housing density [15]. The Centers for Disease Control and Prevention (CDC) suggests that blood Pb testing data may also not sufficiently capture this risk. For example, from 2011 to 2016, in North Carolina, less than a third of children born in the state had BLL screenings at both 1 year and 2 years of age [16]. In rural areas where there are fewer health facilities, BLL screening compliance may be lower. Low blood screening rates combined with the lack of regular water testing pose a significant risk to the health of children in rural areas. Typical Pb sources in water tend to be Pb water service lines and other lead components in plumbing. Pb soldered copper plumbing can contribute to Pb in tap water. Corrosive water increases Pb particulate shedding from the piping and increased particulate Pb. Particulate Pb is associated with higher elevations in BLLs than dissolved Pb [17].

The health effects of elevated BLLs are well known. Children exposed to Pb are known to be more likely to have neurodevelopmental delays, lower scholastic performance, attention deficits, hyperactivity, and increased psychological mood disorders [18,19,20,21]. The effects of chronic exposure to Pb has been shown to negatively affect children’s IQ, and the deficits continue with lower vocabulary and academic performance and a greater risk of not completing high school, which is associated with lower socioeconomic status in the future [22,23]. Bone Pb concentration in children 12–18 is also associated with juvenile delinquency [24]. Even Pb blood levels below the CDC reference value of 3.5 µg/dL have been shown to have effects on IQ and behavior. BLLs as low as 1.2 µg/dL can have effects on IQ [25]. These effects are especially concerning in rural populations which are already at risk for earlier mortality due to low socioeconomic status, and the gap is growing each year compared to urban areas [26]. Chronic Pb exposure can result in reduced socioeconomic opportunity due to poor school performance potentially reducing the ability of children to gain economic opportunities through education. Water Pb exposure above 3.3 μg/L has been shown to significantly increase the odds of a child having an elevated BLL [7].

We have previously reported WLLs in 151 Illinois homes that rely on PDWs for drinking water [27]. In that study, water Pb was detectable in nearly 50% of homes, and 3% of homes had WLLs that exceeded the 15 µg/L action level that applies to community water systems. The primary goal of this study was to assess the likely contribution of drinking water to BLLs among children residing in homes supplied by PDWs in three Illinois counties.

## 2. Materials and Methods

### 2.1. Setting

In this cross-sectional study, two kitchen tap water samples from 151 households in three Illinois counties—Kane, Jackson, and Peoria—were collected by home residents based on the guidelines set by the EPA Lead and Copper Rule [28]. Local health department environmental health staff conducted in-home water sampling training with participants. After at least a 6 h period of no water use, stagnant and flushed (1st and 7th liter, respectively) water samples were returned to the Illinois State Water Survey laboratory to be analyzed for Pb and corrosivity properties. Total Pb was analyzed using graphite furnace atomic absorption. Samples with Pb > 2 µg/L were passed through a 0.45 μm filter and analyzed for dissolved Pb. Sulfate and chloride were measured using ion chromatography and alkalinity was measured by titration. Corrosivity was determined using the Larson–Skold Index (LSI): LSI = (epm chloride + epm sulfate)/(epm alkalinity) where epm = equivalents per million.

### 2.2. Questionnaire

Participants completed a 32-item questionnaire to collect information on demographics and potential Pb exposure risk factors such as plumbing materials, well maintenance, and water treatment. The questionnaire was based on several national surveys and those used in previously published research [29,30,31,32,33,34,35]. The questions address water use, plumbing infrastructure, water treatment devices, primary drinking and cooking water sources, fixtures, plumbing maintenance, housing characteristics, and water safety perceptions. For items about water safety, a 5-point Likert scale was used to assess safety perceptions. The other questions were structured in a multi-choice format or as a yes-or-no question.

### 2.3. Blood Lead Level Modeling

The primary aim of this study was to estimate the contribution of drinking water to BLLs of children (BLLs were not measured). We accomplished this by using the EPA’s Integrated Exposure Uptake Biokinetic (IEUBK) Model Version 2 [36]. IEUBK was developed using data mainly from Pb superfund sites to model risk to children. Geometric mean (GM) BLLs can be modeled using air, dietary, maternal BLL at childbirth, soil and dust, and water Pb concentrations from sites where data are available [36]. Because our interest was in the incremental contribution of PDW water to BLLs, sources and pathways included in IEUBK, except for water exposure, were set to the IEUBK default values (the IEUBK model does not include input terms for paint Pb, other than through a dust Pb exposure pathway). Differences in estimated BLLs were evaluated between the observed 90th and 10th percentile WLLs. We used the IEUBK default values for lead in settled dust, diet, maternal, soil, and air. The only input we changed was WLLs based on the data collected. The aim was to estimate the effect differing WLLs have on BLLs, not directly simulate exact BLLs of children, on PDWs in the three counties. Calculations were based on WLLs from all homes in the study, regardless of whether children lived in these homes or not. For Pb levels below the lab detection limit (<0.76 µg/L), Pb presence was determined using (0.76 µg/L)/(√2), Distributions of BLLs was calculated using IEUBK (version 2) software. Geometric standard deviation (GSD) was at the default value of 1.6, based on the IEUBK user manual guidance. For statistical analysis, STATA (STATACorp LLC, College Station, TX, USA) was used.

## 3. Results

Table 1 shows key characteristics of the three counties. In Jackson County, about 30% of housing is rural, with 60% built before 1980. By comparison, 25% of the housing in Kane County is pre-1980, and the same is true of about half the housing stock in Peoria. The percentage of children under 18 years ranged from 18.4 (Jackson) to 25.0 (Kane) [13]. Details on demographics and the study methodology have been previously published [27].

Table 2 shows Pb concentrations in the 1st liter (stagnant) and 7th liter (flushed) samples. Measurable Pb was present in 73 (48.3%) of the 1st liter samples. Measurable Pb in 5 (3.3%) of 1st liter samples had Pb concentrations >15 µg/L (the EPA action level for community water systems). The highest Pb concentration was 76.2 µg/L, 5 times higher than the EPA action level for community water systems. Five samples had Pb levels above the action level. Measurable Pb was present in 34 (22.5%) of the 7th liter samples, but none had Pb >15 µg/L.

Table 3 shows housing age and presence of detectable Pb. Though a majority of the houses that had Pb present were built prior to 1980, there were homes that were built after 1980 with detectable Pb (27.9% of Pb containing samples). However, out of the homes surveyed, a majority of the pre-1950 and 1970–1979 home water samples contained Pb.

Table 4 shows results from 150 households that returned the questionnaire (99.3% response rate). Several questions detailed participant water usage, treatment equipment, and perceptions of water safety. Most respondents either strongly agreed or somewhat agreed that tap water was safe to drink (75.7%), and the vast majority were not concerned about Pb in water (76.8%). Of the participants who answered the question about the primary water source at home, 31.5% used tap water that was not treated or filtered with any device. The only item with a significant difference across groups was “Amount Spent on Well Upkeep Annually,” where a higher percentage of those with detectable Pb (14.3%) spent greater than $100 per year on well upkeep than those with no detectable Pb (3.9%) (odds ratio 0.25 (95% CI 0.04–1.02)).

Using deciles of 1st liter water Pb concentration, we estimated the WLL effect on geometric mean (GM) BLLs among children aged 6 months to 7 years who were served by a PDW (Table 5). The difference in BLLs between those with a 90th percentile value and a 10th percentile value in water lead level was estimated to be 0.4 µg/dL. Children 0.5 to 2 years of age had the highest modeled GM BLLs. For children in that age category, a difference in estimated BLLs for those in the 90th and the 10th percentile was 0.37 µg/dL.

The IEUBK probability distributions from the deciles are in Figure 1. The cutoff point was set at 3.5 µg/dL, the current CDC reference level for BLLs, with acknowledgement that no Pb level is thought to be entirely safe [37]. Based on the cutoff, the percentage of children at or above the reference value is estimated from all Pb sources in the model. As water WLL inputs increased from 0.54 µg/L to 4.98 µg/L, the probability of a child having a IEUBK-estimated BLL at or above 3.5 µg/dL increased from 18.0 to 27.4%.

## 4. Discussion

This study highlights the risks of Pb exposure among children living in homes with PDWs. Among participants, 31.5% reported that they relied on tap water without any filtration treatment as their source of drinking water, and 62.8% used untreated tap water for cooking. Children who drink untreated tap water or infants who have formula prepared using it are at risk of Pb exposure. Although the observed 90th percentile WLL was below the EPA community system water action level of 15 µg/L, modeled BLLs indicate that children consuming tap water at the 90th percentile value would have BLLs approximately 0.3–0.4 µg/dL greater than those who consume tap water with a WLL level at the 10th percentile value. Therefore, even WLLs below 15 µg/L have the potential to elevate BLLs, which would be expected to impair health. The EPA action level is not a health-based standard.

During the COVID-19 pandemic, rural children in Illinois experienced a 26.9% decrease in BLL testing [38]. Prior to the pandemic, it had already been seen in South Carolina that rural children are at risk of elevated BLLs and that BLLs increased from ages <1 to 2 years of age when screened [39]. With the reduction in testing seen in Illinois, more children are at risk of undetected elevated BLLs.

Other studies have used IEUBK to model BLLs based on water Pb intake. However, inputs for other Pb exposure sources for the IEUBK model vary. A previous study used default inputs for other lead exposures and only altered the WLL variable, like the method used in this current study [40,41]. Deshommes et al. (2013) estimated BLLs in children in relation to WLLs in homes with Pb service lines using the IEUBK model [42]. In that study, WLLs were measured, and model inputs for other Pb sources were either default IEUBK values, Health Canada values, or values from other studies. To confirm the accuracy of the IEUBK models, simulated BLLs were compared to available BLLs from Montreal children from homes with and without Pb service lines. IEUBK estimates were found to be consistent with observed BLLs. For our study, however, BLLs were not measured. Thus, our modeled BLLs could be over- or under-estimates.

Several studies have used Stochastic Human Exposure and Dose Simulation (SHEDS), which is based on a probabilistic Monte Carlo analysis. This method considers how likely the source is to be the main exposure pathway. These exposure values were then used in the IEUBK parameters to model GM BLLs from varying WLL exposures [43,44]. The SHEDS–IEUBK model allows for greater specificity when modeling GM BLLs by predicting the dominant route of exposure and simulating the resulting BLL based on that particular exposure route. With the primary medium of exposure identified, IEUBK-modeled BLLs closely resemble the BLLs seen in National Health and Nutrition Examination Survey data [42]. Zartarian et al. (2017) suggested that SHEDS–IEUBK models should be used when determining costs and benefits of interventions to mitigate Pb [43]. Since the IEUBK models simulated for this study do not take into account other sources and pathways of Pb exposure such as paint Pb, we may be over- or under-estimating the contribution of water Pb to BLLs.

This study had several strengths. Participants had a high rate of compliance with instructions. Participants completed the study questionnaire with a 99.3% response rate. Another strength was the ability to compare people’s self-reported perceptions of their water safety to the actual WLL in their homes. Another strength was that through IEUBK software, changes in WLLs below the action limit set by the EPA Lead and Copper rule were able to be related to the percentage of children whose simulated BLLs were above the 3.5 µg/dL limit. This study estimated that 18% of children will have BLLs above the reference value at Pb levels at the 50th percentile for 1st liter water samples collected during the study. Since participants were able to have health department staff members deliver their sampling kits and demonstrate in their homes correct sampling methods. Additionally, because samples were returned through shipping, participants in remote rural areas were able to participate in the study. Participants who may not have other health services near them were able to provide water samples, which may help capture risk to children living in remote rural areas. It also may capture children who may have missed BLL measurements at 1 and 2 years of age.

This study also had several limitations. Even with site-specific modeling, IEUBK is only an estimate of BLLs. BLLs in combination with housing-specific measurements of Pb levels in paint, dust, and soil likely provide the best analysis of risk. Without these data, we are only able to estimate the effect water levels could have on overall change in BLLs using model default values and not direct effects. Another limitation of our study is sample size. With only 151 homes in three counties, our findings were not representative of these counties or of the state of Illinois. For example, Peoria County has approximately 18,000 PDWs in total, of which we sampled 52. This small sample size, combined with the fact that nearly half of the houses were built after 1980, may have not captured the true risk of Pb in PDWs in these three counties. Filtration systems also may impact the Pb levels, however; we did not collect samples which both included filtration and ones that bypassed filtration. Specific sources of Pb in these water systems was unknown. For example, Pb could be in some combination of the home’s plumbing, the well, or from other parts of the water system.

Future studies of WLLs should include BLLs and other common Pb exposures to accurately assess the risk of Pb in PDWs. As seen in other studies, when more site-specific data or population-specific data are included, IEUBK modeling can be adjusted to provide more accurate BLLs resulting from specific exposures. Since the default IEUBK inputs are largely from superfund sites, the true Pb levels in homes from various sources may have smaller or larger impacts on overall BLLs, based on age of housing, presence of Pb paint, and other exposure factors. Future studies should work to confirm the accuracy of IEUBK default values and the associated simulation of BLLs for homes on PDWs using increased sample sizes and more detailed Pb data for other environmental media. Future studies should involve measuring Pb directly from PDWs and from taps to determine whether the well components or the plumbing is driving WLLs seen in homes, especially since several homes in this study that were built after 1980 had measurable WLLs. The effectiveness of Pb filtration systems should also be considered when taking future water samples.

## 5. Conclusions

The findings suggest that the BLLs of children who consume well water containing measurable Pb are likely e increased as a result of that exposure. The EPA Lead and Copper Rule does not require owners of PDWs to test their drinking water for Pb. Since PDWs are mostly in less dense population areas, children living in the rural United States are particularly at risk of having elevated BLLs due to Pb exposures in water from PDWs. Policies to encourage and financially support testing of tap water in homes with PDWs are needed. Priority should be given to those homes in which children and women of childbearing age reside or are expected to reside, as well as homes built before 1986. Mitigation of Pb in water from PDWs should be adequately funded. Based on our simulation, WLLs that remain below the EPA water Pb action level for community water systems are associated with increased BLLs among young children. Children should not be exposed to Pb in drinking water due to the known adverse health effects at any level.

## Figures and Tables

**Figure 1 ijerph-21-00337-f001:**
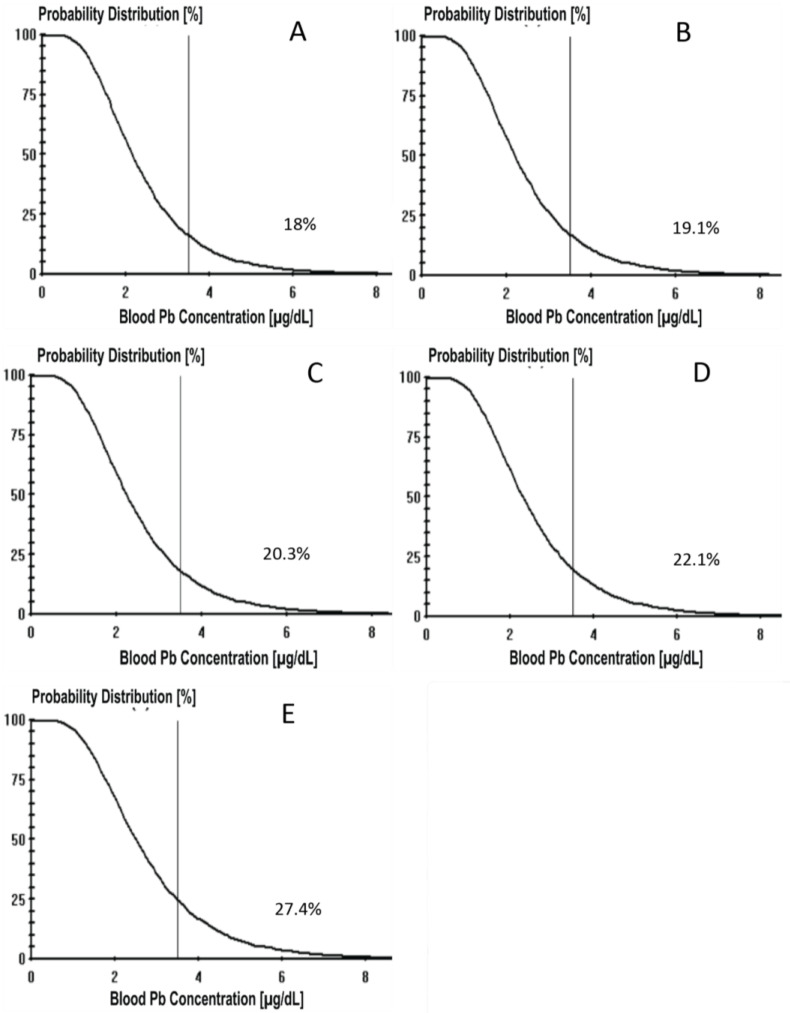
(**A**) IEUBK density curve of BLLs at WLL 0.54 μg/L with the cutoff at 3.5 μg/dL for BLL. (**B**) IEUBK density curve of BLLs at 1.07 WLL μg/L. (**C**) IEUBK density curve of BLLs at WLL 1.65 μg/L. (**D**) IEUBK density curve of BLLs at WLL 2.50 μg/L. (**E**) IEUBK density curve of BLLs at WLL 4.88 μg/L. BLL density curves are for total Pb exposure for all sources (not only WLL). Percentages are the expected % to fall above the cutoff of 3.5.

**Table 1 ijerph-21-00337-t001:** Characteristics of study sample by county.

	Jackson	Kane	Peoria
Population, *n* (%)			
Under 5 years	3129 (5.4)	32,962 (6.2)	12,555 (7.0)
5 to 9 years	2544 (4.4)	32,638 (6.1)	11,115 (6.2)
Under 18 years	10,646 (18.4)	132,979 (25.0)	42,236 (23.6)
Total Population	57,977	532,403	179,179
Rural Housing, %	30	11	17
Housing before 1980, %	60	75	50.5
Estimated PDWs, *n*	1900	7100	18,000

**Table 2 ijerph-21-00337-t002:** Measurable water Pb concentrations by county for 1st and 7th liter samples.

		Water Pb Concentration (µg/L)			
	Samples with Detectable Pb (n)	Minimum Detectable	Maximum	Median	IQR *
1st Liter					
Jackson (n = 38)	20	0.83	76.2	0.85	1.62
Kane (n = 62)	27	0.83	47.0	0.54	1.62
Peoria (n = 51)	26	0.79	15.4	0.79	1.12
All (n = 151)	51	0.79	76.2	0.54	1.60
7th Liter					
Jackson (n = 38)	11	0.89	3.93	0.54	0.46
Kane (n = 62)	8	0.86	3.37	0.54	0
Peoria (n = 51)	15	0.77	5.00	0.54	0.37
All (n = 151)	34	0.77	5.00	0.54	0

* Interquartile range (IQR) is the measure of the spread of the middling values of the data.

**Table 3 ijerph-21-00337-t003:** Housing age and detectable water Pb.

Year House Built	Measurable Pb *n* (%)	Undetectable Pb *n* (%)	Total *n*
Pre-1950	10 (71.4)	4 (28.6)	14
1950–1969	15 (50.0)	15 (50.0)	30
1970–1979	24 (63.2)	14 (36.8)	38
1980–Present	19 (31.7)	41 (68.3)	60

**Table 4 ijerph-21-00337-t004:** Water perception and use between measurable and undetectable water Pb groups.

Question/Answer	Measurable Pb *n* (%)	Undetectable Pb *n* (%)	Total *n* (%)
Amount Spent on Well Upkeep Annually (*n* = 146)			
<$100	60 (85.7)	73 (96.1)	133 (91.1)
>$100	10 (14.3)	3 (3.9)	13 (8.9)
*p* = 0.028 ***			
My tap water is safe to drink (*n* = 148)			
Strongly agree	32 (45.7)	41 (52.6)	73 (49.3)
Agree somewhat	23 (32.9)	16 (20.5)	39 (26.4)
Neither agree nor disagree	10 (14.3)	10 (12.8)	20 (13.5)
Disagree somewhat	4 (5.7)	8 (10.3)	12 (8.1)
Disagree strongly	1 (1.4)	3 (3.8)	4 (2.70)
*p* = 0.370			
Concern about lead in well water after learning about the Flint water crisis (*n* = 138)			
Yes	18 (26.9)	14(19.7)	32 (23.2)
*p* = 0.320			
Primary source of water (*n* = 149)			
Tap Water	25 (35.2)	22 (28.2)	47 (31.5)
Bottled Water	21 (29.6)	20 (25.6)	41 (27.5)
Treated Tap Water	15 (21.1)	26 (33.3)	41 (27.5)
Multiple Sources	3 (4.2)	1 (1.3)	4 (2.7)
Other	7 (9.9)	9 (11.5)	16 (10.7)
*p* = 0.393			
Water Treatment devices (point of use or point of entry) (*n* = 106)			
Reverse osmosis	8 (17.4)	12 (20.0)	20 (18.9)
Ultraviolet light system	0 (0)	1 (1.7)	1 (0.9)
Water Softener	30 (65.2)	37 (61.7)	67 (63.2)
Multiple Devices	8 (17.4)	10 (16.7)	18 (17.0)
*p* = 0.820			
Amount of Tap Water Consumed Daily by Drinking (*n* = 149)			
1–3 cups	12 (16.9)	9 (11.5)	21 (14.1)
4–7 cups	23 (32.4)	26 (33.3)	49 (32.9)
8 or more cups	23 (32.4)	27 (34.6)	50 (33.6)
None	13 (18.3)	16 (20.5)	29 (19.5)
*p* = 0.822			
Cooking Water Source (*n* = 148)			
Unfiltered tap water	46 (64.8)	47 (61.0)	93 (62.8)
Filtered tap water	5 (7.0)	6 (4.11)	11 (7.4)
Bottled water	4 (5.6)	2 (1.0)	6 (4.1)
Other	16 (22.5)	22 (14.99)	38 (25.7)
*p* = 0.688			

Chi-squared test used to calculate *p*-values. * denotes significance at the *p* < 0.05 level.

**Table 5 ijerph-21-00337-t005:** IEUBK Modeled BLLs by age group.

	BLLs in µg/dL by Age Group in Years						
WLL µg/L (WLL Percentile)	0.5–1	1–2	2–3	3–4	4–5	5–6	6–7
0.54 (10–50th)	3.0	3.0	2.4	2.1	2.1	1.9	1.7
1.07 (60th)	3.0	3.0	2.4	2.1	2.1	1.9	1.8
1.65 (70th)	3.1	3.1	2.5	2.2	2.1	2.0	1.8
2.50 (80th)	3.2	3.1	2.5	2.3	2.2	2.0	1.9
4.88 (90th)	3.4	3.3	2.7	2.5	2.4	2.2	2.1

## Data Availability

The data presented in this study are available on request from the corresponding author.

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
