# Peer review of "IEUBK Modeling of Children’s Blood Lead Levels in Homes Served by Private Domestic Wells in Three Illinois Counties"

_ijerph, 2024, doi:10.3390/ijerph21030337_

Round 1

Reviewer 1 Report

Comments and Suggestions for Authors

This study was intended to study the impact of lead in untreated well water on children’s blood lead levels, based on field investigation and IEUBK model. However, there are big problems.

1.       Only water samples were sampled and detected for lead, however, how about other environmental media? Since the children’s blood lead levels could be assessed based on the main environmental media such as food, drinking water, soil, air and so on, via dermal contact, ingestion and inhalation exposure pathways by IEUBK model, the evaluation of blood lead levels just according to drinking water was not reliable and scientific.

2.       The contents were too little to support a scientific paper.

3.       The useful data in Table 3 was little. In other words, a lot of useless information was illustrated in Table 3.

4.       The analysis and discussions on the potential influencing factors shown in Table 3, such as water treatment devices, water intake, on children’s blood lead levels and lead pollution in well water should be done.

5.       The discussion between lead-contained well water ingestion and children’s blood lead levels were not deep and enough.

Comments on the Quality of English Language

Some sentences were a little bit long, and the expression could be improved.

Reviewer 2 Report

Comments and Suggestions for Authors

The study assesses health risk scenarios based on lead concentration in private domestic well water where water quality is not regularly assessed. in three Illinois counties.

Ln 22 in abstract The highest concentration evaluated was 76.2 μg/L, concentration significantly higher than EPA action level (15 μg/L). How many samples were higher than 15 μg/L?

The first parts of the introduction were prepared carelessly, for example:

Ln 28 The abbreviation WWLs was not previously defined. 

Ln 43 the sentence is incomplete.

Ln 54 Use abbreviation of PDW

Ln 62 Provide the reference where it is indicated that 15 μgPb/L in water  represents action level that applies to community water system 

It is recommended to facilitate the understanding of the results in the evaluation of lead in water and blood samples the following:

Concentrations of lead in water reported in μg/L(ppb)

Blood lead concentrations in μg/dL

The IEUBK probability distributions model.  It appears to have been well applied and is able to denote how the contribution of lead in water can increase the risk of increasing the concentration of lead in blood.

Very good presentation of the strengths and limitations of the study

Reviewer 3 Report

Comments and Suggestions for Authors

This was a cross sectional investigation of well water lead levels in three counties within Illinois and modeled blood lead levels.  It is well executed and utilized a well-known method of using IEUBK modeling to estimate blood lead levels from known water lead.  The authors found that use of this application estimated that a significant number of children would result in having blood lead levels above 3.5 ug/dl of blood lead, which is a significant number and a public health issue.  The study is very centered on the fact that there are a significant number of people within these three counties that have well water and there is concern in general resulting from the Flint Michigan story that there are potential exposures from lead in water exposures.

However, water is often not the whole story, houses, particularly older houses; they indicate pre 1980 , but census data cites pre 1950 and pre 1970 data , shows an incredibly high risk of increased risk of blood lead poisoning for children for both the lead in the paint that imparts risk , but also the leaded pipes in these homes that is totally independent of well water but is a significant risk factor for childhood blood lead poisoning.  This paper will be much more relevant if they include the literature on pre fifty and pre 70 housing and blood lead levels and risk of increased blood lead levels.   

They should stratify their modeling data on this variable and present the differences in levels and expanding their literature to include older housing by stratifying by age of housing and presenting stratified data or a multiple regression with this as a controlling variable. Below are some references I believe will help them to make the paper more well-rounded and inclusive and also more relevant. See references in file below 

Reviewer 4 Report

Comments and Suggestions for Authors

In this paper, the main objective was to assess the contribution of drinking water to blood lead level for children, by using IEUBK model. The study is really interesting, well written and well presented/structured. I have some minor comments attached in the pdf version.

Round 2

Reviewer 1 Report

Comments and Suggestions for Authors

The manuscript has been revised according to the suggestions.

Comments on the Quality of English Language

The language of the manuscript could be revised before under the consideration of acceptance.

Reviewer 3 Report

Comments and Suggestions for Authors

numerous grammatical and spelling errors ,  I have marked them throughout , needs someone to read this line by line  to remove them ,  

I think the paper is vastly improved and they indicated that obviously older housing is a paramount risk factor over and above water making this a public health issue as children are a large portion of the inhabitants in these three counties,  I commend their work.  Given these issues, I hope there will be followup with blood lead levels to corroborate the remediation effects.  

Comments on the Quality of English Language

 Numerous typos and  sentence structure, I flagged as many as I saw but needs to be gone over with an editor . 

 I think the paper is vastly improved and they indicated that obviously older housing is a paramount risk factor over and above water making this a public health issue as children are a large portion of the inhabitants in these three counties, I commend their work. 

Given these issues, I hope there will be follow-up with blood lead levels to corroborate the remediation effects.  
